# Factors associated with severe sepsis in diarrheal adults and their outcome at an urban hospital, Bangladesh: A retrospective analysis

**Monira Sarmin** (ID), **Monjory Begum** (ID), **Farhana Islam**, **Farzana Afroze** (ID), **Lubaba Shahrin**, **Sharifuzzaman**, **Tahmina Alam**, **Abu Sadat Mohammad Sayeem Bin Shahid** (ID), **Tahmeed Ahmed**, **Mohammod Jobayer Chisti** (ID) *

Nutrition and Clinical Services Division (NCSD), International Centre for Diarhoeal Disease Research, Bangladesh (icddr,b), Dhaka, Bangladesh

* chisti@icddrb.org

## Abstract

**Data Availability Statement:** The Data contain personal information of the patients and therefore cannot be made publicly available. Data are

### Background

To describe factors associated with severe sepsis in diarrheal adults and their outcomes and offender in blood and stool to understand their interplay as clinical features of sepsis and severe diarrhea often overlap.

### Methods and results

We used this retrospective chart analysis employing an unmatched case-control design to study critically ill diarrheal adults aged ≥18 years treated in ICU of Dhaka hospital, icddr,b between January 2011 to December 2015. Of 8,863 in-patient diarrheal adults, 350 having severe sepsis were cases and an equal number of randomly selected non-septic patients were the controls. Cases died significantly more (14.9% vs 4.6%, p = <0.001) than controls. 69% of the cases progressed to septic shock. In logistic regression analysis, steroid intake, ileus, acute kidney injury (AKI), metabolic acidosis, and hypocalcemia were significantly associated with severe sepsis in diarrheal adults (all, p<0.05). 12% of cases (40/335) had bacteremia. *Streptococcus pneumoniae* [9 (22.5%)] was the single most common pathogen and gram-negatives [27 (67.5%)] were prevailing as a group.

### Conclusion

Diarrheal adults who had ileus, AKI, metabolic acidosis, hypocalcemia, and also took steroids were found to have an association with severe sepsis. Strikingly, gram-negative were the predominant bacteria among the diarrheal adults having severe sepsis.

available upon request from the IRB, icddr,b for researchers who meet the criteria for access to confidential data.The part of the data set related to this manuscript is available upon request and readers may contact the head of the Research Administration (Ms Armana Ahmed, aahmed@icddrb.org) of icddr,b (http://www.icddrb.org/).

**Funding:** The author(s) received no specific funding for this work.

**Competing interests:** The authors have declared that no competing interests exist.

# Introduction

Sepsis with its unpredicted cryptic course can affect any patient anywhere in the world. Sepsis results from a disrupted inflammatory response to infection in the body where it can progress to septic shock followed by multiple organ dysfunction syndrome and death [1]. Worldwide, the burden of sepsis is perceived to be high, though it remained understudied as 87% of the world population residing in lower-middle-income countries. A study including data from 1979 to 2015 projected a global burden of annual 31.5 million sepsis, and 19.4 million severe sepsis, and among them, 5.3 million died [2]. Studies from Asian critical care facility including Bangladesh reported poor compliance of the surviving sepsis campaign (SSC) bundles and the case-fatality rate was as high as 49·2% from severe sepsis [3, 4].

Dhaka hospital of icddr,b treats diarrheal adults, some of them also have severe sepsis and often experience fatal outcomes. Patients typically present with features of sepsis evident by infection, tachycardia, fever, and leukocytosis. Progression of sepsis resulting in hypotension and/or absent peripheral pulses from poor peripheral perfusion in the absence of dehydration is termed severe sepsis. Unresponsiveness to isotonic fluid (30 ml/kg bolus of normal saline/ ringer's lactate over 10–15 min) and require the support of inotropes/vasopressors, is termed as septic shock. Oliguria, acute kidney injury, and altered mental status signify the presence of organ dysfunction [5, 6]. Diarrheal pathogens may translocate from gut to blood stream and cause sepsis. Several literatures presented this diarrhea, sepsis interplay mostly in children from Bangladesh [7, 8]. Following SSC recommendation, we administer the first antibiotic early which helps to reduce in-hospital mortality [9].

Thus, we need to predict severe sepsis at its early stage and prompt management of severe sepsis, which may help to reduce deaths by preventing the progression of severe sepsis to septic shock. However, data are lacking in diarrheal adults having sepsis, or consequences of sepsis in adults hospitalized for diarrhea. Thereby, we aimed to portray the clinical findings with the outcome of severe sepsis in diarrheal adults comparing them with non-septic adults and also describe the spectrum and frequency of offending pathogens in blood and stool.

# Methods

## Study design

We retrospectively analyzed data and for that, we included diarrheal patients aged ≥18 years who were treated for diarrhea in icddr,b Dhaka hospital between January 2011 to December 2015, where cases had severe sepsis along with diarrheal illness and the controls had only diarrhea. For analysis, we followed an unmatched case-control design. We had excluded patients having cardiogenic shock, and hospital-acquired severe sepsis. Infection or the presumed presence of infection plus tachycardia plus hyperthermia (≥38.5˚C) or hypothermia (≤35.0˚C), or abnormal white blood cell numbers are the criteria of sepsis [10, 11]. A combination of sepsis and poor peripheral perfusion evident by hypotension and/or absent peripheral pulses without dehydration constituted severe sepsis [10]. Usually, in the absence or after correction of dehydration, if a patient with sepsis remained hemodynamically unstable (mean arterial pressure <65 mm Hg or urine output <0.5 ml/kg/hr) we labeled that as severe sepsis in that patient.

Fluid (bolus of 30 ml/kg normal saline/ringer's lactate over 10–15 min) unresponsiveness and requirement of inotropes confirmed the presence of septic shock. We compared the clinical characteristics of diarrheal adults with and without severe sepsis. Every year, clinicians at Dhaka hospital manage approximately 150,000 patients having diarrhea, with or without other complications and more than 90% were managed in the short stay ward without performing any laboratory investigations. Usually, within a year there is a peak during winter mostly of

Rotavirus affecting children and during pre-monsoon, mostly of *Vibrio Cholerae*, E. *Coli* affecting adults.

Among them, 8863 adults required either admission to longer stay unit (LSU) or Intensive care unit (ICU) for their sickness and thus constituted our sampling frame. From these 8863, for this unmatched case-control study, we have selected 350 cases and an equal number of controls were chosen for comparison at random using a statistical package for the social sciences [IBM SPSS Statistics for Windows, Version 20.0. Armonk, NY: IBM Corp]. We used their data for this chart analysis. A description of the Dhaka hospital has been illustrated in other papers [12].

ICU physicians re-evaluated patients having severe sepsis, started the required workup, and prescribed a standard management plan following the hospital's guidelines. A portable pulse oximeter (OxiMaxN-600) measured capillary oxygen saturation (SpO$_2$) and Accu Chek Active (*Roche Diagnostics* GmbH, *Mannheim*, *Germany*) glucometer estimated blood glucose. Hypoxemic patients received oxygen therapy (@4–5 liter/minute). In alignment with SSC 2012 guideline, hospital protocol has been developed. Before instituting antibiotics (within one hour of diagnosis) e.g., 3$^{rd}$ generation cephalosporin and aminoglycosides (fluoroquinolone instead of aminoglycosides for concomitant community-acquired pneumonia), blood was drawn for culture and sensitivity and other investigations. Appropriate feeding (nothing by mouth with maintenance fluid for patients with septic shock), was provided as and when required. They received 30 ml/kg fluid bolus (normal saline/ringer's lactate) and also received vasopressor and inotropes for septic shock evident by unresponsiveness to a fluid bolus. We started noradrenaline first, 0.05 microgram/kg.min and increased the dose after 15 minutes to 0.1 microgram/kg.min to a maximum of 0.5 microgram/kg.min. We used definitive response criteria: mean arterial pressure (MAP) $\geq$ 65 mm Hg, urine output (UO) $\geq$0.5mL/kg/hour, and central venous pressure (CVP) 8 to 12 mm Hg [13, 14]. As we have limited expertise, we used to measure CVP in the external jugular vein, which is reliable, instead of the internal jugular vein [15]. Thereafter, we added injection adrenaline if the goal was not achieved. In inotrope-resistant shock, we also added injection hydrocortisone. Monitoring of the heart rate, respiratory rate, skin color/capillay refill time (CRT), temperature, pulse oximetry, and mental status [16] were ensured besides the aforementioned criteria.

## Data management

Dhaka hospital, icddr,b provides a unique number for every patient, and against this number, all the data are recorded. Our research team accessed the databases to obtain the retrospective data used in our study during the year 2016–2017. We have collected the data from a computer-based record-keeping network, SHEBA. We prepared the case report forms (CRF) and finalized them for data accession after pretesting.

**Participant recruitment method.** Patient identification numbers were collected from the data repository, then designated physicians with prior approval accessed individual patient data set to review their information and selected participants following inclusion and exclusion criteria. A paper-based CRF was first filled up and thereafter the data were transferred to a soft copy for analysis and record keeping.

**Information collected.** We collected demographics (age, gender), clinical information (presence of fever, cough, respiratory distress, disorientation, and their duration, history of [H/O] antibiotic use for the current illness, systemic steroid intake, and comorbidities e.g., chronic lung disease, diabetes mellitus, abnormal auscultatory findings in the lungs, and severe sepsis), and laboratory data (blood, and stool cultures and sensitivity, total white blood cell, serum creatinine, serum electrolytes, calcium, and magnesium) from the medical chart. We

presented laboratory data as total leucocyte count (white blood cell count 4.0–11.0 x10^9/L) hyponatremia (serum sodium <135 mmol/L), hypokalemia (serum potassium <3.5 mmol/L), hypocalcemia (serum total calcium <2.12 mmol/L), hypomagnesemia (serum total magnesium <0.65 mmol/L), hypoglycemia (random blood sugar <3.0 mmol/L), metabolic acidosis (serum $TCO_2$ <24 mmol/L) and acute kidney injury (AKI) if serum creatinine was 1.5 times the upper limit of normal, [Normal serum creatinine is 159 μmol/L in men and 146 μmol/L in women]. Ileus was defined as a clinical condition where a patient had abdominal pain with or without vomiting and sluggish/absent bowel sound for which the patient has deferred any oral diet.

Invasive diarrhea is defined as diarrhea where stool is mucoid, small in volume, with or without visible blood, and presented with fever, tenesmus/cramping abdominal pain. The presence of pus cell >20/HPF and RBC (any number) in stool with alkaline pH also supports the diagnosis of invasive diarrhea.

Three or more watery stools in a day is termed as acute watery diarrhea (AWD). Patients having AWD may present with no dehydration, some or severe dehydration. We categorized some and severe dehydration as dehydrating diarrhea.

We also documented the hospital course and outcome and the number of days in the hospital.

## Data analysis

We entered all the data into SPSS for Windows and Epi Info (version 7.0, Epi Info™ software; Center for Disease Control and Prevention, Atlanta, GA, USA). We reported categorical data, like numbers and percentages; continuous data as means with standard deviations or medians with interquartile ranges (IQRs) as appropriate. The Student's t-test compared means of homogenous data and the Mann-Whitney U test compared inhomogeneous data between the groups. Odds ratio (OR) and their 95% confidence intervals (CIs) were used to demonstrate the strength of association. For statistically significant, a p-value is set <0.05. To identify factors for severe sepsis in diarrheal adults, initially, a bivariate model was used, and then a multivariable logistic regression analysis model identified factors independently associated with severe sepsis after controlling for the relevant confounding variables.

## Ethical considerations

In this study, we reviewed only the medical records without involving any interviews with patients or caregivers. Data were anonymized before analysis. Nevertheless, the Institutional Review Board of icddr,b comprised of two bodies named as Research Review Committee (RRC) and Ethics Review Committee (ERC) and both RRC and ERC approved the study.

## Results

During the study period, out of 8,863 diarrheal in-patients at Dhaka hospital, icddr,b, 350 were the cases and thus, 350 were controls. So, the percentage of severe sepsis is 3.95 (350/8863*100) in diarrheal adults.

Cases had significantly higher mortality than controls (Table 1).

Admission, the median age (years) (45 Vs 50) was comparable between the groups. Systemic steroid intake before this illness (23% Vs 2%), disorientation (25% Vs 6%), ileus (56% Vs 38%), and pneumonia (58% Vs 19%) were significantly higher among the cases compared to controls (Table 1). The cases had more hypoglycemia (9% Vs 0.3%), AKI (66% Vs 34%), metabolic acidosis (90% Vs 70%), hypomagnesemia (74% Vs 49%), and hypocalcemia (87% Vs 55%) and less often had hypokalemia (29% Vs 38%) (Table 2) and more often required long (hours) (68 Vs 48) hospital stay (Table 1) compared to the controls. Among the cases, 240

**Table 1. Clinical characteristics of diarrheal patients with and without severe sepsis.**

| Characteristics | Severe sepsis | | OR (95% CI) | P |
|---|---|---|---|---|
| | Yes, n = 350 (%) | No, n = 350 (%) | | |
| Age (years) (median, IQR) | 45 (30, 60) | 50 (30, 65) | - | 0.054 |
| Male sex | 189 (54) | 191 (55) | 1.0 (0.72–1.31) | 0.94 |
| Complaints for fever | 237 (68) | 153 (44) | 2.7 (1.99–3.67) | <0.001 |
| Presence of cough | 58 (17) | 43 (12) | 1.42 (0.93–2.17) | 0.132 |
| Presence of respiratory distress | 115 (33) | 58 (17) | 2.46 (1.72–3.53) | <0.001 |
| Duration of diarrhea (days) (Median, IQR) | 1 (1, 2) | 2 (1, 3) | - | <0.001 |
| Duration of fever (days) (Median, IQR) | 2 (1, 3) | 3 (2, 6) | - | <0.001 |
| Duration of cough (days) (Median, IQR) | 2 (1.75,5.5) | 3 (2, 7) | - | 0.189 |
| Disorientation | 86 (25) | 22 (6) | 4.86 (2.96–7.97) | <0.001 |
| Prior antibiotics use | 30 (9) | 44 (13) | 0.65 (0.36–1.06) | 0.110 |
| Systemic steroid intake | 80 (23) | 6 (2) | 17 (7.3–39.5) | <0.001 |
| Co-morbidity | 111 (32) | 115 (33) | 0.94 (0.69–1.30) | 0.808 |
| Hypoxemia | 89 (25) | 24 (7) | 4.39 (2.72–7.09) | <0.001 |
| Dehydrating diarrhea* | 149 (42.6) | 125 (35.7) | 1.33 (0.98–1.81) | 0.074 |
| Hypoglycemia | 27 (9) | 1 (0.3) | 29.48 (3.98–218.44) | <0.001 |
| Adventitious lung sounds | 148 (42) | 63 (18) | 3.34 (2.36–4.71) | <0.001 |
| Pneumonia | 203 (58) | 67 (19) | 5.91 (4.20–8.32) | <0.001 |
| Ileus | 195 (55.7) | 132 (37.7) | 2.07 (1.54–2.81) | <0.001 |
| Septic shock | 240 (69) | 0 | undefined | <0.001 |
| Hospital stay (hours) (Median, IQR) | 68 (25, 103) | 48 (24,72) | - | <0.001 |
| Referral# | 129 (37) | 81 (23) | 1.94 (1.39–2.70) | <0.001 |
| Deaths | 33/221 (14.9) | 12/269 (4.46) | 3.7 (1.89–7.47) | <0.001 |

* Diarrhea with some or severe dehydration.

# In severe sepsis, predominant causes of referral were renal and cardiac: (50+42 = 92).

(69%) patients developed septic shock and required inotropes (Table 1). Regarding the comorbidities among cases, 20.9%, 12%, and 6.3% had chronic lung diseases, Diabetes Mellitus, and arthritis respectively.

**Table 2. Laboratory characteristics of diarrheal patients with and without severe sepsis.**

| Characteristics | Severe sepsis | | OR (95% CI) | P |
|---|---|---|---|---|
| | Yes, n = 350 (%) | No, n = 350 (%) | | |
| Hb (gm/dl) (mean ± SD) | 12.13±2.59 | 12.3 ±2.74 | - | 0.344 |
| Total leukocyte count (10^9/L) (Median, IQR) | 14.25 (9.39, 19.89) | 11 (7.47, 15.03) | - | <0.001 |
| Hyponatremia | 94 (27) | 74 (30) | 0.87 (0.61–1.25) | 0.511 |
| Metabolic acidosis | 306 (90) | 171 (70) | 3.63 (2.33–5.64) | <0.001 |
| Hypokalemia | 94 (29) | 90 (38) | 0.66 (0.47–0.95) | 0.030 |
| AKI* | 223 (66) | 80 (34) | 3.75 (2.64–5.32) | <0.001 |
| Hypocalcemia | 274 (87) | 35 (55) | 5.34 (3.06–10.00) | <0.001 |
| Hypomagnesemia | 229 (74) | 29 (49) | 2.99 (1.69–5.31) | <0.001 |
| Invasive diarrhea | 120 (62) | 133 (64) | 0.94 (0.63–1.41) | 0.842 |
| Blood isolates | 40/335 (12) | 27/ (18) | 0.63 (0.37–1.07) | 0.113 |
| Stool isolates | 22 (21) | 26 (23) | 0.89 (0.47–1.68) | 0.838 |

*AKI = Acute kidney injury (if serum creatinine is 1.5 times the upper limit of normal).

**Table 3. Bacterial isolates from the blood of the study patients.**

| Blood isolates | Severe sepsis | |
|---|---|---|
| | Yes, n = 40 (%) | No, n = 27 (%) |
| *S. pneumoniae*\* | 9 (22.5) | 0 |
| *S.* Typhi | 4 (10) | 14 (52) |
| *S. paratyphi* | 2 (5) | 0 |
| *Pseudomonas* spp | 7 (17.5) | 2 (7) |
| *E. coli* | 2 (5) | 5 (18) |
| *Enterobacter* spp | 6 (15) | 1 (4) |
| *Klebsiella* spp | 3 (7.5) | 0 |
| *Enterococcus* spp | 2 (5) | 1 (4) |
| *Acinetobacte*r spp | 1 (2.5) | 3 (11) |
| *S. aureus*\* | 4 (10) | 1 (4) |

\* *S. pneumoniae* and *S. aureus* are gram-positive (32.5%) and the rest are gram-negatives in the severe sepsis group.

The blood and stool isolate among the cases and the controls are shown in Tables 3 and 4 respectively. Among the cases, only 12% had bacteremia (40/335). As a single pathogen *Streptococcus pneumoniae* (22.5%) was the predominant bacterial isolate whereas as a group, gram negatives (67.5%) were prevailing (Table 3). The percentage of *Shigella*, *Salmonella*, and *Vibrio* in stool were comparable between the groups.

In logistic regression analysis, systemic steroid intake, ileus, AKI, metabolic acidosis, and hypocalcemia were independently associated with severe sepsis (Table 5).

## Discussion

We studied diarrheal adults to explore clinical and laboratory factors associated with severe sepsis and their outcome. We have also evaluated the spectrum and frequency of bacteremia in this population. Overall, the predominance of gram negatives among diarrheal adults having severe sepsis is the striking observation of this study. Other important observations of our study were: (i) progression to septic shock from severe sepsis in diarrheal adults was high (69% [240/350]), (ii) Mortality rate was significantly higher among patients having severe sepsis and diarrhea compared to those without severe sepsis/sepsis, (iii) systemic steroid intake, ileus, AKI, metabolic acidosis, and hypocalcemia were independently associated with severe sepsis.

The predominance of gram negatives among the bacterial pathogens isolated from diarrheal adults having severe sepsis is remarkable. We may speculate that in diarrheal adults' breach of healthy intestinal flora might allow potential translocation of gram-negative pathogens to the bloodstream. Except for severe sepsis by *Streptococcus pneumoniae* from a respiratory source, gram negatives were found to be common pathogens in several previous studies

**Table 4. Bacterial isolates from the stool of the study patients.**

| Stool isolates | Severe sepsis | |
|---|---|---|
| | Yes, n = 22 (%) | No, n = 26 (%) |
| *Shigella* spp | 8 (36) | 7 (27) |
| Salmonella group | 4 (18) | 8 (31) |
| *Campylobacter* | 2 (9) | 3 (11.5) |
| *Vibrio* | 6 (27) | 5 (19) |
| *Aeromonas* | 2 (9) | 3 (11.5) |

**Table 5. Logistic regression (backward conditional) analysis to explore the clinical predictors of septic shock.**

| Characteristics | OR | CI | P |
|---|---|---|---|
| Systemic steroid intake | 8.87 | 1.55–30.41 | 0.011 |
| Disorientation | 1.20 | 0.53–2.72 | 0.664 |
| Hypoglycemia | 3.48 | 0.41–29.74 | 0.255 |
| Pneumonia | 1.55 | 0.74–3.25 | 0.241 |
| Ileus | 2.95 | 1.43–6.09 | 0.003 |
| Metabolic acidosis | 2.83 | 1.10–7.29 | 0.031 |
| AKI | 2.14 | 1.03–4.42 | 0.041 |
| Hypocalcemia | 8.20 | 3.79–17.75 | <0.001 |
| Hypokalemia | 0.53 | 0.26–1.08 | 0.080 |
| Hypomagnesemia | 1.56 | 0.70–3.52 | 0.279 |

OR = odds ratio, CI = confidence interval, AKI = Acute kidney injury.

involving adults with sepsis/severe sepsis and conducted in Southeast Asia and Europe [17, 18].

Using the diagnostic criteria suggested by SSC guidelines, we found the proportion of septic shock among adults having severe sepsis was 69%, which is consistent with a recent study [19]. When diarrheal patients developed severe sepsis, their risk of death increases significantly. Among ICU patients, it is the 2[nd] leading cause of death [18]. The mortality rate for sepsis in different studies varies and could be as high as 56% [18, 20]. The deleterious cascade of sepsis causes multiorgan failure and death. Thus, the relationship between increased mortality and septic shock is well established [21, 22]. Our observation of 14.9% mortality among diarrheal adults with severe sepsis might reflect good adherence to SSC guidelines. On the other hand, 129 cases and 81 controls required referral to different facilities for other associated illnesses as we did not have the facilities to treat those. Although we did not know their outcome, it is postulated that the mortality rate among them might also be higher.

We observed patients with a H/O systemic glucocorticoid intake had a significantly higher risk of severe sepsis when they also had diarrhea. Glucocorticoids are alluring drugs with their beneficial role for endocrine, inflammatory, allergic, and immunological disorders [23]. Besides, they increase the probability of infection with common bacteria, viruses, and fungus by impairing phagocyte function [24, 25] and producing hypogammaglobulinemia even with short courses from outpatient settings [26]. Our observation of the association of ileus with severe sepsis is also understandable. Bacteria may translocate into the systemic circulation from an insulted intestine that may cause sepsis [27]. Diversely, during sepsis, splanchnic hypoperfusion and gut-derived mediators induced leukocytes malfunction hamper gastrointestinal motility and result in paralytic ileus [28].

We also observed a significant number of patients who suffered from AKI. 66% of our study patients with severe sepsis developed AKI, which is consistent with previous observations [29, 30]. Septic AKI is a consequence of the renal microvascular dysfunction, the interaction between pathogen fragments and renal cells, cytokine storm, or disintegration between injured organs [31]. However, after the restoration of the target mean arterial pressure (MAP), most of our study patients recovered from AKI, only 50 out of 223 required referral to the renal specialized hospital due to the persistence of AKI. Studies also describe that a portion of patients without renal replacement therapy (RRT) recovered spontaneously with optimal sepsis care [32, 33]. The findings of our study support and greatly extend those of previous findings. Metabolic acidosis is also anticipated in severe sepsis. Severe sepsis can produce

metabolic acidosis in a variety of ways. Either insufficient oxygen delivery with tissue hypoxia and resultant anaerobic glycolysis [34] or overproduction of pyruvate [35] may lead to lactic acidosis in severe sepsis. Septic AKI makes the kidney unable to excrete waste products of nitrogen metabolism (urea and creatinine) which also contributes to acidosis [36]. Again, some to severe dehydration can cause AKI by causing renal hypoperfusion, however, it gets better with appropriate hydration of a patient having only diarrhea. Though recommended by SSC guidelines, in our settings, we could not check the serum lactate level due to economic constraints which might help to differentiate between sepsis and dehydration.

Calcium in its active form (ionized Ca) can diffuse across cellular membranes to maintain body homeostasis [37]. Hypocalcemia, in association with severe sepsis [38], is commonly found in hospitalized critically ill patients [39]. In our study, we observed that hypocalcemic patients were at higher risk of having severe sepsis. Here, we have the facilities to measure total calcium and we used to treat hypocalcemic patients with intravenous infusion of calcium gluconate followed by oral supplementation of calcium carbonate. Contrasting evidence exist, about the role of calcium supplementation in severe sepsis. A Cochrane review concluded that supplementation has no added benefit, [40] whereas a recent retrospective study demonstrated calcium supplementation has a protective role for 28-days mortality [41].

## Study limitations

The main limitation of our study was that we were unable to perform a follow-up of patients who were referred for management of myocardial infarction and renal failure and thus their outcome was not known to us. This might have an impact on having only 14.9% mortality among our study population with severe sepsis, which is lower than those in other studies. A comparison between patients having "sepsis plus diarrhea" and "only sepsis" might help us to get insight how diarrhea influences sepsis. However, as this is a diarrhea treatment facility and all of our patients have diarrhea, we were unable to compare them with non-diarrheal adults having sepsis. As the data captured information from 2011–2015 where we did not follow the qSOFA or NEWS, we were unable to report qSOFA or NEWS which are an important aspect for early diagnosis of sepsis. On the other hand, we also have some strengths of our study. It was a five-year follow-up study involving a good number of patients. Dhaka hospital of icddr,b has a standard protocol that follows SSC guidelines. So, the quality of care did not differ much. Although retrospective observation, this is the first study where we have evaluated severe sepsis among adults who were hospitalized for diarrhea.

## Generalizability

Our sample is a part of a larger population visiting icddr,b Dhaka hospital each year, however, they mainly presented with diarrheal illness, thus may be generalizable among hospitalized diarrheal adults.

## Conclusion

Our data suggest that severe sepsis (3.95%) is common in diarrheal adults and the rate of progression from severe sepsis to septic shock is high (69%). Cases had a higher case-fatality rate than the controls. Systemic steroid intake, ileus, AKI, metabolic acidosis, and hypocalcemia were independently associated with severe sepsis in diarrheal adults. In resource-poor settings like ours, proper history taking, rigorous follow-ups, and early identification of organ dysfunction are essential for a better outcome.

## Acknowledgments

We gratefully acknowledge the core donors for their support and commitment to icddr,b's research efforts. We would like to express our sincere thanks to all clinical fellows, nurses, members of the feeding team, and cleaners of the hospital for their invaluable support and contribution to patient care.

## Author Contributions

**Conceptualization:** Monira Sarmin, Monjory Begum, Farhana Islam, Farzana Afroze, Lubaba Shahrin, Sharifuzzaman, Tahmina Alam, Abu Sadat Mohammad Sayeem Bin Shahid, Tahmeed Ahmed, Mohammod Jobayer Chisti.

**Data curation:** Monira Sarmin, Monjory Begum, Farhana Islam, Tahmina Alam.

**Formal analysis:** Monira Sarmin, Monjory Begum, Sharifuzzaman, Mohammod Jobayer Chisti.

**Methodology:** Monira Sarmin, Abu Sadat Mohammad Sayeem Bin Shahid, Mohammod Jobayer Chisti.

**Supervision:** Mohammod Jobayer Chisti.

**Writing – original draft:** Monira Sarmin.

**Writing – review & editing:** Monira Sarmin, Monjory Begum, Farhana Islam, Farzana Afroze, Lubaba Shahrin, Sharifuzzaman, Tahmina Alam, Abu Sadat Mohammad Sayeem Bin Shahid, Tahmeed Ahmed, Mohammod Jobayer Chisti.

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
