## [Decision Letter · Decision Letter 0]

20 May 2021

PONE-D-21-06147

Factors associated with severe sepsis in diarrheal adults and their outcome at an urban hospital, Bangladesh: A retrospective analysis

PLOS ONE

Dear Dr. Chisti,

Thank you for submitting your manuscript to PLOS ONE. After careful consideration, we feel that it has merit but does not fully meet PLOS ONE’s publication criteria as it currently stands. Therefore, we invite you to submit a revised version of the manuscript that addresses the points raised during the review process.

We look forward to receiving your revised manuscript.

Kind regards,

Chiara Lazzeri

Academic Editor

PLOS ONE

Journal Requirements:

"In this study, we reviewed only the medical records without involving any interviews with patients or caregivers. Data were anonymized before analysis. Nevertheless, the Institutional Review Board of icddr,b approved the study."

3. Thank you for providing the date(s) when patient medical information was initially recorded. Please also include the date(s) on which your research team accessed the databases/records to obtain the retrospective data used in your study.

4. In your Methods section, please provide additional information about the participant recruitment method and the demographic details of your participants. Please ensure you have provided sufficient details to replicate the analyses such as:

a) a description of any inclusion/exclusion criteria that were applied to participant selection,

b) a statement as to whether your sample can be considered representative of a larger population

5.  Thank you for stating the following in theFunding Section of your manuscript:

"This work was supported by icddr,b and its donors which provide unrestricted support

to the institution for its operations and research. Current donors providing unrestricted support

include the Governments of Bangladesh, Canada, Sweden, and the UK."

6. We note that you have indicated that data from this study are available upon request. PLOS only allows data to be available upon request if there are legal or ethical restrictions on sharing data publicly. For more information on unacceptable data access restrictions, please see http://journals.plos.org/plosone/s/data-availability#loc-unacceptable-data-access-restrictions.

7. Your abstract cannot contain citations. Please only include citations in the body text of the manuscript, and ensure that they remain in ascending numerical order on first mention.

Reviewers' comments:

Reviewer's Responses to Questions

**Comments to the Author**

1. Is the manuscript technically sound, and do the data support the conclusions?

Reviewer #1: No

Reviewer #2: Yes

Reviewer #3: Partly

Reviewer #4: Yes

2. Has the statistical analysis been performed appropriately and rigorously? 

Reviewer #1: I Don't Know

Reviewer #2: Yes

Reviewer #3: No

Reviewer #4: I Don't Know

3. Have the authors made all data underlying the findings in their manuscript fully available?

Reviewer #1: Yes

Reviewer #2: Yes

Reviewer #3: Yes

Reviewer #4: Yes

4. Is the manuscript presented in an intelligible fashion and written in standard English?

Reviewer #1: Yes

Reviewer #2: No

Reviewer #3: No

Reviewer #4: Yes

5. Review Comments to the Author

Reviewer #1: 1/ The standard and widely accepted criteria/definition for sepsis should be used.

2016 SCCM/ESICM criteria of sepsis- "Sepsis is defined as life-threatening organ dysfunction caused by a dysregulated host response to infection. Organ dysfunction can be identified as an acute change in total SOFA score ≥2 points consequent to the infection."

Neither tachycardia, temperature or WBC are a part of the this criteria. They were part of the old definition but were removed because these findings could be see in other conditions too (like- diarrhea with dehydration!). Hence, it was not a reliable way to define sepsis.

Reference no. 1 of their study uses 2016 SCCM/ESICM definition of sepsis as well. So, they should use this definition too as it will differentiate sepsis vs. diarrhea with severe dehydration in a more reliable way. Or the authors should describe how they reliably differentiated between the two as both cases can have tachycardia, increased WBC.

2/ " Diarrheal adults who had ileus, AKI, metabolic acidosis, hypocalcemia, and also took steroids were prone to have severe sepsis." The prevalence of factors like AKI, ileus, metabolic acidosis was different between cases (sepsis+ diarrhea) and control (diarrhea) as a consequence of sepsis. The presence of these factors did not make the cases more likely to develop sepsis.

3/ How were cases and controls matched? What was done to account for confounders? What is the rationale for the choice of cases and controls? How was the sample size arrived at?

More importantly- why was case-control design chosen? The aim- "to portray the clinical findings with the outcome of severe sepsis in diarrheal adults comparing them with non-septic adults" with diarrhea- is suited for a cohort study, either retrospective or prospective.

In fact, "mortality", "progression to septic shock" is being compared between the 2 groups in the study- this is already a retrospective cohort approach.

4/ All outcomes, exposures, predictors, potential confounders, potential effect modifiers should be defined before the study was performed. The research question is very vague, and outcomes have not been defined before the study was undertaken. Most of the outcomes seem to be arrived at post-hoc.

5/ Introduction- what is the current knowledge, including some recent relevant articles, regarding the topic? what is the gap in the current knowledge and how does this study aim to fill the gap? There are already tools like qSOFA, NEWS to "to predict severe sepsis at its early stage"

6/ "The predominance of gram negatives among the bacterial pathogens isolated from diarrheal adults having severe sepsis is remarkable. We may speculate that in diarrheal adults’ breaches of healthy intestinal flora might allow potential translocation of gram-negative pathogens to the bloodstream". It is more remarkable that S. pneumonia was the most common isolated organism in diarrheal patients with sepsis as it is not an enteric pathogen or commensal.

Also the authors write a lot about bacterial isolates in Results and Discussion, when this was not the objective of the study in the first place.

7/ I suggest a thorough review of grammar and English by a native speaker as there are many errors.

8/ Many references are more than 10 years old. 4 of them are more than 30 years old

A study regarding diarrhea and sepsis from such a large hospital specialized in diarrhea will be valuable. However, I suggest a complete revision of the study design and a more thoughtful and precise objective of the study to be clearly defined first.

Reviewer #2: The authors investigated factors to predict severe sepsis at its early stage for prompt management through a retrospective case-control analysis.

1/The authors stated that out of 350 patients with severe sepsis, 149 patients (50%, which should be 42.6%) had diarrhea with some or severe dehydration. However, the criteria authors used to diagnosis sepsis are consistent with older definitions of sepsis in Fethi et al study which were (Temperature >38°C or <36°C, tachycardia, respiratory rate, and WBCs)

Bacterial diarrhea with severe dehydration can be clinically overlapped with severe sepsis however, the author stated that “evident by hypotension and/or absent peripheral pulses without dehydration constituted severe sepsis”. Thus, I think authors should explain and clarify which criteria they used to diagnose 149 patients with dehydration as severe sepsis which could be overlapped with their definition of severe sepsis.

Gül, Fethi et al. “Changing Definitions of Sepsis.” Turkish journal of anaesthesiology and reanimation vol. 45,3 (2017): 129-138. doi:10.5152/TJAR.2017.93753

2/In results, authors mentioned that “On admission, the cases had lower mean age, H/O systemic steroid intake before this illness, disorientation, ileus, pneumonia (Table 1)” However, in table 1, cases were higher in the above mentions events except for age. Also, it is not reported how many patients with ileus for both groups.

I recommend to re-write the whole paragraph in a more organized manner outlining the percentage of the most important features with more interpretation for odd ratios and their statistically significant value. Also, a summary sentence for stool isolates.

3/ Were the diagnosis of pneumonia and AKI prior to diarrhea or on admission or as a consequence in all patients?

4/ In the methods section, the authors mentioned that they presented leukocytosis as laboratory data, however, table 2 shows only levels of Hb. Mean values of WBCs are important to diagnose sepsis as mentioned in the authors’ criteria to diagnosis sepsis.

5/ General format, paraphrasing, and English grammar need to be improved and revised in the manuscript

Reviewer #3: 1. Abstract: Words are repeated in 2 consecutive sentences.

Rephrasing the writing abstract is recommended in a good Journal.

2. Introduction:

How the author should clarify the sepsis and severe diaphysis often overlap? The evidences. Thus, we need to predict severe sepsis at its early stage and prompt management of severe sepsis will reduce deaths in these adults by eliminating its progression to septic shock -> The sentence should be concise.

“Thus, we need to predict severe sepsis at its early stage and prompt management of severe sepsis, which will reduce deaths by preventing progression to septic shock.”

The author needs to add to the causes why diarrheal adults having sepsis should be careful, not just missing data. Because this will strongly support why the author did this study.

3. Discussion

L196-198: Some patients require transfers to different facilities -> so the authors should report the exact number of cases and controls moved cause clinical outcomes may be altered.

Rephrasing the writing discussion is recommended in a good Journal. The paragraphs should be organized.

4. Conclusion

L246-247: How many patients with diarrheal have severe sepsis? What is the ratio?

The authors wrote that severe sepsis is common in diarrheal adults and the rate of progression from severe sepsis to septic shock is high -> but there is no number, rate,.. to support it.

As the data are in the results “L146: 8,863 diarrheal inpatients at Dhaka hospital, icddr,b, 350 were the cases…” -> I can understand what this number is 350/8863*100= 3.95% severe sepsis in diarrheal.

In conclusion, I would like to find recommendations such as study findings in diarrheal patients with severe sepsis features -> clinicians should examine and detect early sepsis case.

Reviewer #4: Dear author,

The topic is really interesting, in that it could improve the treatment of septic patients.

1/ The introduction insists on the importance to conduct studies on septic patients which was transmitted through the statistics and mortality /cost rates. The problem is why should we study them in ''diarrheal patients'' - emphasize on diarrheal profile of patients and why it is important with some statistics preferably from your country .

2/ The choice of ''diarrhea+sepsis'' VS ''only diarrhea'' is questionable since we can foresee the differences , maybe a ''sepsis +diarrhea'' VS ''only sepsis'' approach would prove more beneficial in order to validate your aim which is improving the treatment of septic patients - please discuss this points in discussion , look for references if there are studies conducted with that comparison and add this point in the limits of your study

3/ Please justify the sample size of 350 , why that exactly ? was it estimated before ?

4/ Why that exact time period ? , was there some cold epidemics or other infectious epidemics in the country during the 5 year study period ? which could explain the pneumonia /respiratory septic patients rate ? if so discuss that hypothesis in the discussion section

5/ study setup:

the hospital receives ''150.000 diarrhea cases'' then later you said only ''8863 patients '' were found , lastly you said ''only 350 fulfilled the criteria ''

 please recheck the statistics of the hospital and rectify if there was a mismatching in the manuscript

- Please write a clear apart section in method for ''inclusion criteria '' so that we can understand why only 350 from 8863 were eligible .

6/ Table 1 ''comorbidity'' please define that , like how many diabetes (which could be linked to sepsis ), cancer/ immuno suppression (also can cause sepsis), smoking(which could cause an increase in respiratory sepsis) /alcohol status,

cardiac problems, renal insufficiency (which could allow us to predict the referral rate)

7/ I would appreciate it if you can add the ''origin of the sepsis'' like from where did the infection start just to get an idea of most frequent sources

8/Tables 2 : invasive diarrhea  explain this variable how was that defined , is it different form dehydrating diarrhea mentioned in table 1

9/ Table1 death rate , i understood that it was the rate from those not referred patients but the table makes it seem like it was from the total 350  please mention the total in the death row

10/ L212 while discussing the AKI causes i think mentioning the dehydration as a potential cause of sepsis which could be caused also by fever/inflammatory process is worth it.  please add that to the discussion

11/ L159 bacteriemia rate is 40/335 while in table 2 it is 40/350 please rectify that

12/ L152/L167 you talk all over the manuscript about ileus association with sepsis but the problem is that i could not find this factor in table 1 to which you refer- please rectify that and add ileus to table 1

13/ For logistic regression analysis, you opted for ''backward conditional '' model , why is that particular model which is very ''mathematical why not the simpler ''Enter'' model ?? were other models tested before ?

14/ Another remark please justify your sepsis definition and reference it.

6. PLOS authors have the option to publish the peer review history of their article (what does this mean?). If published, this will include your full peer review and any attached files.

Reviewer #1: No

Reviewer #2: **Yes: **Nourin Ali Sherif

Reviewer #3: **Yes: **Dang Thi Phuong Dung

Reviewer #4: **Yes: **Nacir Dhouibi

---

## [Author Response · Author response to Decision Letter 0]

18 Jul 2021

Responses to the comments of the academic editor in bold

Journal Requirements:

 Response: Thank you for notifying this. We have tried to follow the recommendations.

"In this study, we reviewed only the medical records without involving any interviews with patients or caregivers. Data were anonymized before analysis. Nevertheless, the Institutional Review Board of icddr,b approved the study."

Response: Thank you for your suggestion. We revised the ethics statement-"In this study, we reviewed only the medical records without involving any interviews with patients or caregivers. Data were anonymized before analysis. Nevertheless, the Institutional Review Board (IRB) of icddr,b comprised of two bodies named as Research Review Committee (RRC) and Ethics Review Committee (ERC) and both RRC and ERC approved the study." (page 8)

3. Thank you for providing the date(s) when patient medical information was initially recorded. Please also include the date(s) on which your research team accessed the databases/records to obtain the retrospective data used in your study.

 Response: Thank you for your query. Our research team accessed the databases/records to obtain the retrospective data used in our study during the year 2016-2017 (added this on page 6 under ‘data management section’)

 4. In your Methods section, please provide additional information about the participant recruitment method and the demographic details of your participants. 

Response: Thank you for your suggestion. 

Participant recruitment method: Patient`s identification numbers were collected from the data repository, then designated physicians with prior approval, accessed individual patient data set to review their information, and selected participants depending on inclusion ad exclusion criteria. A paper-based CRF was first filled up and thereafter the data were transferred to a soft copy for analysis and record keeping. (added in the manuscript on page 6, 7 under ‘patient recruitment method’ section.)

Demographics details: We are sorry that except for age and gender, other demographic data were not recorded properly. Thus, we are not able to include detail demographic data in the table.

Please ensure you have provided sufficient details to replicate the analyses such as:

a) a description of any inclusion/exclusion criteria that were applied to participant selection,

b) a statement as to whether your sample can be considered representative of a larger population

Response:

Thank you for the suggestion.

Inclusion criteria:

1. Diarrheal adults aged ≥18 years

2. Severe sepsis (case group)

3. Non-septic adults (control group)

Exclusion criteria:

1. Cardiogenic shock and hospital-acquired severe sepsis

We have mentioned these on page 4 under ‘study design’.

Our sample is a part of a larger population visiting icddr,b Dhaka hospital each year, however, they are mainly admitted with diarrheal illness and may be representative of diarrheal population. (We have included this on page 18 under ‘Generalizability’ section) 

5. Thank you for stating the following in the Funding Section of your manuscript:

"This work was supported by icddr,b and its donors which provide unrestricted support

to the institution for its operations and research. Current donors providing unrestricted support

include the Governments of Bangladesh, Canada, Sweden, and the UK."

 Response: Thank you. We have followed the instructions. Revised the funding statement and added it in the rebuttal letter.

6. We note that you have indicated that data from this study are available upon request. PLOS only allows data to be available upon request if there are legal or ethical restrictions on sharing data publicly. For more information on unacceptable data access restrictions, please see http://journals.plos.org/plosone/s/data-availability#loc-unacceptable-data-access-restrictions. 

Response: Regarding data availability, it is prudent to mention that the data of this manuscript have been obtained from a large data set that deals with several objectives! The submitted manuscript deals with one of the objectives of that data set! This data set contains some personal information of the study patients (such as name, admission date, month, area of residence) those were required during ensuring follow-up of the patients. However, it has been ensured to our IRB that the personal information of the patients will not be disclosed, but, the study results will be published. Thus, the availability of this whole data set in the manuscript, the supplemental files, or a public repository will open all the personal information of the patients those should not be disclosed; additionally, this will disclose other important information those are yet to be published! Thus, the policy of our centre (icddr,b) is that the part of data (de-identified) set related to this manuscript is available upon request and readers may contact the head of the Research Administration (Ms Armana Ahmed, aahmed@icddrb.org) of icddr,b (http://www.icddrb.org/) We have mentioned this in our rebuttal letter. 

7. Your abstract cannot contain citations. Please only include citations in the body text of the manuscript, and ensure that they remain in ascending numerical order on first mention.

 Response: Thank you. We have rechecked the abstract for consistency. We have also checked the citations in the body text of the manuscript, and they remained in ascending numerical order on the first mention.

Reviewers' comments:

Reviewer's Responses to Questions 

Comments to the Author

1. Is the manuscript technically sound, and do the data support the conclusions?

Reviewer #1: No

Reviewer #2: Yes

Reviewer #3: Partly

Reviewer #4: Yes

2. Has the statistical analysis been performed appropriately and rigorously? 

Reviewer #1: I Don't Know

Reviewer #2: Yes

Reviewer #3: No

Reviewer #4: I Don't Know

3. Have the authors made all data underlying the findings in their manuscript fully available?

Reviewer #1: Yes

Reviewer #2: Yes

Reviewer #3: Yes

Reviewer #4: Yes

4. Is the manuscript presented in an intelligible fashion and written in standard English?

Reviewer #1: Yes

Reviewer #2: No

Reviewer #3: No

Reviewer #4: Yes

5. Review Comments to the Author

Reviewer #1: 1/ The standard and widely accepted criteria/definition for sepsis should be used.

2016 SCCM/ESICM criteria of sepsis- "Sepsis is defined as life-threatening organ dysfunction caused by a dysregulated host response to infection. Organ dysfunction can be identified as an acute change in total SOFA score ≥2 points consequent to the infection."

Neither tachycardia, temperature or WBC are a part of the this criteria. They were part of the old definition but were removed because these findings could be see in other conditions too (like- diarrhea with dehydration!). Hence, it was not a reliable way to define sepsis.

Reference no. 1 of their study uses 2016 SCCM/ESICM definition of sepsis as well. So, they should use this definition too as it will differentiate sepsis vs. diarrhea with severe dehydration in a more reliable way. Or the authors should describe how they reliably differentiated between the two as both cases can have tachycardia, increased WBC.

Response: Thank you so much for your suggestions. 2016 SCCM/ESICM definition of sepsis is a revised one that is assumed to differentiate sepsis reliably. However, we were analyzing data back from 2011 to 2015, so, we used older criteria for the diagnosis of severe sepsis. 

After correction of some or severe dehydration, if a septic patient remained hemodynamically unstable (mean arterial pressure <65 mm hg or urine output <0.5 ml/kg/hr) we labeled that severe sepsis in that patient (mentioned this on page 4 under ‘study design’)

2/ " Diarrheal adults who had ileus, AKI, metabolic acidosis, hypocalcemia, and also took steroids were prone to have severe sepsis." The prevalence of factors like AKI, ileus, metabolic acidosis was different between cases (sepsis+ diarrhea) and control (diarrhea) as a consequence of sepsis. The presence of these factors did not make the cases more likely to develop sepsis.

Response: Thanks for your valuable comments. We have revised this sentence in the abstract on page 2. 

3/ How were cases and controls matched? What was done to account for confounders? What is the rationale for the choice of cases and controls? How was the sample size arrived at?

More importantly- why was case-control design chosen? The aim- "to portray the clinical findings with the outcome of severe sepsis in diarrheal adults comparing them with non-septic adults" with diarrhea- is suited for a cohort study, either retrospective or prospective.

In fact, "mortality", "progression to septic shock" is being compared between the 2 groups in the study- this is already a retrospective cohort approach.

Response: Thank you. We have employed an unmatched case-control design. Multivariable logistic regression was done to control the confounder. We wanted to explore the associated factors for severe sepsis, so we chose diarrheal adults with severe sepsis as cases and selected an equal number of controls from diarrheal adults only. This is a time-bound study covering a five-years period. Thus, we included all eligible participants in this study. 

We have chosen an unmatched case control design as we have done it retrospectively. We have identified the outcome (severe sepsis) and later we identified the exposure retrospectively to be associated with severe sepsis. (We have mentioned this on page 5 under ‘study design’)

4/ All outcomes, exposures, predictors, potential confounders, potential effect modifiers should be defined before the study was performed. The research question is very vague, and outcomes have not been defined before the study was undertaken. Most of the outcomes seem to be arrived at post-hoc.

Response: Thank you, this is a retrospective chart analysis. It seems that most of the outcomes are post hoc, however, here outcome variable is severe sepsis, Exposures/predictors variables are systemic steroid intake, disorientation, hypoglycemia, pneumonia, ileus, metabolic acidosis, AKI, hypocalcemia, hypokalemia, hypomagnesemia. Confounders are disorientation, hypoglycemia, pneumonia, AKI, hypokalemia, hypomagnesemia. We do not find any effect modifier as there was no variable that influences the association greatly. 

5/ Introduction- what is the current knowledge, including some recent relevant articles, regarding the topic? what is the gap in the current knowledge and how does this study aim to fill the gap? There are already tools like qSOFA, NEWS to "to predict severe sepsis at its early stage"

Response: Thank you. As our data back from 2011 to 2015, we are unable to provide data on qSOFA or NEWS. We appreciate that these are important in sepsis diagnosis. We have addressed this in the limitation section. (page 17)

6/ "The predominance of gram negatives among the bacterial pathogens isolated from diarrheal adults having severe sepsis is remarkable. We may speculate that in diarrheal adults’ breaches of healthy intestinal flora might allow potential translocation of gram-negative pathogens to the bloodstream". It is more remarkable that S. pneumonia was the most common isolated organism in diarrheal patients with sepsis as it is not an enteric pathogen or commensal.

Also the authors write a lot about bacterial isolates in Results and Discussion, when this was not the objective of the study in the first place.

Response: Thank you for your suggestion. Isolation of pathogens guides the clinician to choose appropriate antibiotic in a septic patient. As a group gram negatives are on the rise. We want to share this information. So, we have revised and included this as one of the objectives. In the last paragraph of the introduction (page 4)

7/ I suggest a thorough review of grammar and English by a native speaker as there are many errors.

Response: Thank you for your suggestion. We have revised the ENGLISH meticulously.

8/ Many references are more than 10 years old. 4 of them are more than 30 years old

A study regarding diarrhea and sepsis from such a large hospital specialized in diarrhea will be valuable. However, I suggest a complete revision of the study design and a more thoughtful and precise objective of the study to be clearly defined first.

Response: Thank you for your valuable and overall positive comment. We have revised the manuscript following your valuable suggestions and hope to be kindly considered for publication. 

Reviewer #2: The authors investigated factors to predict severe sepsis at its early stage for prompt management through a retrospective case-control analysis.

1/The authors stated that out of 350 patients with severe sepsis, 149 patients (50%, which should be 42.6%) had diarrhea with some or severe dehydration. However, the criteria authors used to diagnosis sepsis are consistent with older definitions of sepsis in Fethi et al study which were (Temperature >38°C or <36°C, tachycardia, respiratory rate, and WBCs)

Bacterial diarrhea with severe dehydration can be clinically overlapped with severe sepsis however, the author stated that “evident by hypotension and/or absent peripheral pulses without dehydration constituted severe sepsis”. Thus, I think authors should explain and clarify which criteria they used to diagnose 149 patients with dehydration as severe sepsis which could be overlapped with their definition of severe sepsis.

Gül, Fethi et al. “Changing Definitions of Sepsis.” Turkish journal of anaesthesiology and reanimation vol. 45,3 (2017): 129-138. doi:10.5152/TJAR.2017.93753

Response: Thank you for your suggestion and the article that compared different definitions of sepsis evolved with time.

As we were analyzing data back from 2011 to 2015 where we followed the old criteria, we used older criteria for diagnosis of severe sepsis. 

The reviewer rightly identified that severe dehydration and severe sepsis may overlap. Usually, after correction of dehydration, if a patient with sepsis remained hemodynamically unstable (mean arterial pressure <65 mm hg or UO<0.5 ml/kg/hour) we labeled that as severe sepsis for that patient. After correction of dehydration, 149 patients did not achieve the targeted MAP of >65 mm hg, so, we diagnosed them as having severe sepsis. (on page 4 under ‘study design’ section). We have also corrected the percentage of dehydrating diarrhea in table 1.

2/In results, authors mentioned that “On admission, the cases had lower mean age, H/O systemic steroid intake before this illness, disorientation, ileus, pneumonia (Table 1)” However, in table 1, cases were higher in the above mentions events except for age. Also, it is not reported how many patients with ileus for both groups.

I recommend to re-write the whole paragraph in a more organized manner outlining the percentage of the most important features with more interpretation for odd ratios and their statistically significant value. Also, a summary sentence for stool isolates.

Response: Thank you for your suggestion. Following your recommendation, we have revised the result section. Now we have presented age as median which showed no significant difference. We have added ileus in table 1, added the percentage of the most important features. (revised from page 9-13)

3/ Were the diagnosis of pneumonia and AKI prior to diarrhea or on admission or as a consequence in all patients?

Response: Thank you for the query, pneumonia and AKI were diagnosed at admission based on clinical and laboratory findings done on admission.

4/ In the methods section, the authors mentioned that they presented leukocytosis as laboratory data, however, table 2 shows only levels of Hb. Mean values of WBCs are important to diagnose sepsis as mentioned in the authors’ criteria to diagnosis sepsis.

Response: Thank you for your suggestions, now, we have incorporated a variable `Total Leucocyte count` in table 2. (page 11)

5/ General format, paraphrasing, and English grammar need to be improved and revised in the manuscript

Response: Thank you for your suggestion. We have revised the manuscript following a recommendation from our respected reviewers.

Reviewer #3: 1. Abstract: Words are repeated in 2 consecutive sentences.

Rephrasing the writing abstract is recommended in a good Journal.

Response: Thank you for your suggestion. We have revised the abstract. (page 2)

2. Introduction:

How the author should clarify the sepsis and severe diaphysis often overlap? The evidences. Thus, we need to predict severe sepsis at its early stage and prompt management of severe sepsis will reduce deaths in these adults by eliminating its progression to septic shock -> The sentence should be concise.

“Thus, we need to predict severe sepsis at its early stage and prompt management of severe sepsis, which will reduce deaths by preventing progression to septic shock.”

The author needs to add to the causes why diarrheal adults having sepsis should be careful, not just missing data. Because this will strongly support why the author did this study.

Response: Thank you for guiding us. We have followed the instruction and incorporated your suggested write-up. (page 4)

3. Discussion

L196-198: Some patients require transfers to different facilities -> so the authors should report the exact number of cases and controls moved cause clinical outcomes may be altered.

Rephrasing the writing discussion is recommended in a good Journal. The paragraphs should be organized.

Response: Thank you for the query. 129 cases and 81 controls required referral to different facilities for other associated illnesses. We tried to organize the section. (page 15)

4. Conclusion

L246-247: How many patients with diarrheal have severe sepsis? What is the ratio?

The authors wrote that severe sepsis is common in diarrheal adults and the rate of progression from severe sepsis to septic shock is high -> but there is no number, rate,.. to support it.

As the data are in the results “L146: 8,863 diarrheal inpatients at Dhaka hospital, icddr,b, 350 were the cases…” -> I can understand what this number is 350/8863*100= 3.95% severe sepsis in diarrheal.

In conclusion, I would like to find recommendations such as study findings in diarrheal patients with severe sepsis features -> clinicians should examine and detect early sepsis case.

Response: Thank you. You have rightly identified the percentages. Now we have added this in the manuscript (page 9-Results and 18-Conclusion)

Reviewer #4: Dear author,

The topic is really interesting, in that it could improve the treatment of septic patients.

1/ The introduction insists on the importance to conduct studies on septic patients which was transmitted through the statistics and mortality /cost rates. The problem is why should we study them in ''diarrheal patients'' - emphasize on diarrheal profile of patients and why it is important with some statistics preferably from your country.

Response: Thank you for your suggestion.

Diarrheal adults are also at risk of developing sepsis. Diarrheal pathogens may translocate from gut to blood stream and cause sepsis. Literatures presented this diarrhea, sepsis interplay mostly in children from Bangladesh. (page 3)

2/ The choice of ''diarrhea+sepsis'' VS ''only diarrhea'' is questionable since we can foresee the differences , maybe a ''sepsis +diarrhea'' VS ''only sepsis'' approach would prove more beneficial in order to validate your aim which is improving the treatment of septic patients - please discuss this points in discussion , look for references if there are studies conducted with that comparison and add this point in the limits of your study

Response: Thank you. The proposed approach would be nicer to compare between ''sepsis +diarrhea'' VS ''only sepsis''. However, as this is diarrhea treatment facility and all of our patients have diarrhea, we were unable to compare them with non-diarrheal adults having sepsis. We have added this in our limitation section. (page 17, 18)

3/ Please justify the sample size of 350, why that exactly ? was it estimated before ?

Response: Thank you. We reviewed the existing data and found that 350 diarrheal adults had severe sepsis during the study period. It was not estimated before.

4/ Why that exact time period ? , was there some cold epidemics or other infectious epidemics in the country during the 5 year study period ? which could explain the pneumonia /respiratory septic patients rate ? if so discuss that hypothesis in the discussion section

Response: Thank you so much. We have selected the five-year study period randomly. As per our observation, the number of patients visit for diarrhea was not unusual for the period. Usually, within a year, there is a peak during Winter mostly of Rotavirus affecting children and during pre-monsoon, mostly of Vibrio Cholerae, E. Coli affecting adults. (page 5)

5/ study setup:

the hospital receives ''150.000 diarrhea cases'' then later you said only ''8863 patients '' were found , lastly you said ''only 350 fulfilled the criteria ''

 please recheck the statistics of the hospital and rectify if there was a mismatching in the manuscript

- Please write a clear apart section in method for ''inclusion criteria '' so that we can understand why only 350 from 8863 were eligible .

Response: Thank you for the suggestion. The hospital receives ''150,000 diarrhea cases'' a year is quite correct. Most of them require in-hospital stay for 1-2days for whom minimum medications were used and they required no investigations. However, these 8863 adults required either admission to LSU or ICU for their sickness and thus constituted our sampling frame. From these 8863, we have selected 350 cases following inclusion and exclusion criteria. (page 5)

6/ Table 1 ''comorbidity'' please define that, like how many diabetes (which could be linked to sepsis), cancer/ immuno suppression (also can cause sepsis), smoking (which could cause an increase in respiratory sepsis) /alcohol status,

cardiac problems, renal insufficiency (which could allow us to predict the referral rate)

Response: Thank you for your query.

Regarding the comorbidities among cases, 20.9%, 12%, and 6.3% had chronic lung diseases, Diabetes Mellitus, and arthritis respectively (data not shown in the table). We have no record of smoking.

7/ I would appreciate it if you can add the ''origin of the sepsis'' like from where did the infection start just to get an idea of most frequent sources

Response: Thank you. Here, pathogens causing diarrhea and pneumonia might act as the source of sepsis. In diarrhea, translocation of gut pathogen through the damaged intestinal wall might enter into systemic circulation and causes sepsis syndrome. (page 16)

8/Tables 2 : invasive diarrhea  explain this variable how was that defined , is it different form dehydrating diarrhea mentioned in table 1

Response: Thank you so much. You are right. According to WHO classification, invasive diarrhea and dehydrating diarrhea are two different entities. In invasive diarrhea, usually, there is mucoid, small volume, blood mixed stool with presence fever, tenesmus/cramping abdominal pain. Stool routine examination may show pus cell and RBC with alkaline ph.

Three or more watery stools in day is termed as acute watery diarrhea (AWD). Patients having AWD may present with no dehydration, some or severe dehydration. We categorized some and severe dehydration as dehydrating diarrhea. (page 7, 8)

9/ Table1 death rate , i understood that it was the rate from those not referred patients but the table makes it seem like it was from the total 350  please mention the total in the death row

Response: Thank you. We have added the statistics regarding deaths in the table 1

10/ L212 while discussing the AKI causes i think mentioning the dehydration as a potential cause of sepsis which could be caused also by fever/inflammatory process is worth it.  please add that to the discussion

Response: Thank you for your suggestion. In addition to sepsis, some to severe dehydration can cause AKI by causing renal hypoperfusion, however, it gets better with appropriate hydration of a patient having only diarrhea. We have added this in our discussion. (page 16)

11/ L159 bacteriemia rate is 40/335 while in table 2 it is 40/350 please rectify that

Response: Thank you for the query. We have now corrected the statistics in the table 2.

12/ L152/L167 you talk all over the manuscript about ileus association with sepsis but the problem is that i could not find this factor in table 1 to which you refer- please rectify that and add ileus to table 1

Response: We appreciate your suggestion. Now we have added variable ileus in table 1.

13/ For logistic regression analysis, you opted for ''backward conditional '' model , why is that particular model which is very ''mathematical why not the simpler ''Enter'' model ?? were other models tested before ?

Response: Thank you. We used all the variables significant in the bivariate analysis and then gradual elimination of variables from the regression model to find a suitable model. We have tested Enter method that did not differ much. However, the Backward conditional model is more appropriate to find associated factors with a specific condition, here severe sepsis.

14/ Another remark please justify your sepsis definition and reference it.

Response: Thank you so much. We followed surviving sepsis campaign (SSC) guideline. Infection or the presumed presence of infection plus tachycardia plus hyperthermia (≥38.5°C) or hypothermia (≤35.0°C), or abnormal white blood cell numbers are the criteria of sepsis. We have added a reference for it. (page 5)

6. PLOS authors have the option to publish the peer review history of their article (what does this mean?). If published, this will include your full peer review and any attached files.

Do you want your identity to be public for this peer review? For information about this choice, including consent withdrawal, please see our Privacy Policy.

Reviewer #1: No

Reviewer #2: Yes: Nourin Ali Sherif

Reviewer #3: Yes: Dang Thi Phuong Dung

Reviewer #4: Yes: Nacir Dhouibi

---

## [Editor Report · Decision Letter 1]

7 Sep 2021

Factors associated with severe sepsis in diarrheal adults and their outcome at an urban hospital, Bangladesh: A retrospective analysis

PONE-D-21-06147R1

Dear Dr. Chisti,

We’re pleased to inform you that your manuscript has been judged scientifically suitable for publication and will be formally accepted for publication once it meets all outstanding technical requirements.

Kind regards,

Chiara Lazzeri

Academic Editor

PLOS ONE
---

## [Editor Report · Acceptance letter]

10 Sep 2021

PONE-D-21-06147R1 

Factors associated with severe sepsis in diarrheal adults and their outcome at an urban hospital, Bangladesh: A retrospective analysis 

Dear Dr. Chisti:

I'm pleased to inform you that your manuscript has been deemed suitable for publication in PLOS ONE. Congratulations! Your manuscript is now with our production department. 

Kind regards, 

on behalf of

Dr. Chiara Lazzeri 

Academic Editor

PLOS ONE